# Decolorization and Biodegradability Enhancement of Synthetic Batik Wastewater Containing Reactive Black 5 and Reactive Orange 16 by Ozonation

**Ahsin Pramugani** [1,2,3], **Toshiyuki Shimizu** [1,4], **Shinpei Goto** [1,5], **Teti Armiati Argo** [2] **and Satoshi Soda** [1,*]

1   Department of Civil and Environmental Engineering, Ritsumeikan University, 1-1-1 Nojihigashi, Kusatsu, Shiga 525-8577, Japan
2   School of Architecture, Planning and Policy Development, Institute Technology Bandung, Jl Ganesa 10, Bandung 40132, Indonesia
3   Directorate General of Human Settlements, Ministry of Public Works and Housing, Republic of Indonesia, Jl Pattimura No 20, Jakarta 12110, Indonesia
4   Faculty of Urban Management, Fukuyama City University, 2-19-1 Minato-Machi, Fukuyama, Hiroshima 721-0964, Japan
5   Sewage Division, Nihon Suido Consultants Co. Ltd., 6-22-1 Nishishinjyuku, Shinjyuku, Tokyo 163-1122, Japan
*   Correspondence: soda@fc.ritsumei.ac.jp

**Abstract:** The batik industry generates large amounts of highly colored wastewater. Azo dyes in batik wastewater can cause environmental pollution. In this study, synthetic batik wastewater containing 32 mg/L Reactive Black 5 (RB5) and 32 mg/L Reactive Orange 16 (RO16) was treated by ozonation in a 2 L batch reactor. The wastewater color unit was reduced from 4240 to 70 after 10 min ozonation and to below 50 after 15 min ozonation (7.3 g $O_3$/m$^3$, 4 L/min). The first-order decay constant for 5 min ozonation was determined to be 1.11 min$^{-1}$ for RB5 and 0.82 min$^{-1}$ for RO16. Biodegradation tests using activated-sludge microorganisms showed the toxicity of RB5 and RO16 for microbial respiration and revealed the detoxifican of the dyes by ozonation. Three-dimensional fluorescence spectroscopy analysis indicated the temporal accumulation of ozonolysis products of RO16 and RB5. The chemical oxygen demand concentration of the wastewater was reduced from 86 mg/L to 73 mg/L by biodegradation alone, 63 mg/L by ozonation alone, and 54 mg/L by ozonation followed by biodegradation. Existing wastewater treatment plants using conventional bioprocesses can be upgraded to achieve robust dye treatment by installing the ozonation process as a pretreatment.

**Keywords:** azo dye; batik; biodegradation; decolorization; ozonation

## 1. Introduction

Batik, a technique of wax-resist dyeing applied to whole cloth in South and Southeast Asia [1], has shown increased production since the designation of Indonesian batik as a UNESCO Intangible Cultural Heritage. Consequently, Indonesia is confronted today by severe environmental problems, especially the deterioration of river water quality, caused by the waste and wastewater generated by the batik industry [2]. The batik industry generates large quantities of highly colored wastewater containing dyes, wax, and resin [3–5]. Only small amounts of batik wastewater are treated properly in Indonesia. Our earlier report described the current situations at three wastewater treatment plants (WWTPs) serving the batik industry in Pekalongan City [6]. These WWTPs mainly use biological processes such as activated sludge, anaerobic ponds, and constructed wetlands because of their low operational costs.

However, batik dyes are intentionally designed to be recalcitrant under the usual conditions of usage. In fact, the WWTPs in Pekalongan City have not always conformed to effluent quality standards for biochemical oxygen demand (BOD, 60 mg/L) and chemical oxygen demand (COD, 150 mg/L) [6]. Many studies have confirmed that incomplete

degradation products of dyes can lead to carcinogenic and mutagenic effects in natural environments [7]. Azo dyes containing one or more azo bonds (-N=N-) as chromophores are the most widely used synthetic dyes. Wastewater treatment systems using biological treatments alone are limited in regard to the risk management of hazardous wastewater derived from the growing batik industry. Therefore, combining biological processes and physicochemical processes could be necessary for developing robust alternative treatment systems for batik wastewater.

Physicochemical processes such as baffle separation [3], membrane filtration [5], and hydrodynamic cavitation have previously been studied for batik wastewater treatment [8]. According to a review of batik industry wastewater treatment systems [1], ozonation has not been fully studied. Ozonation is an advanced oxidation process (AOP) used for treating recalcitrant organics and the generation of hydroxyl radicals (•OH), which are powerful oxidizing agents. Ozonation is recognized as a sludge-free method, although ozone generation entails high energy consumption. Chromophore groups with conjugated double bonds, which are responsible for coloring, can be broken down either directly or indirectly by ozone. They form smaller molecules, thereby decreasing the color of the effluents. Similarly to wastewater from other textile processes [9], batik wastewater ozonation is an efficient pretreatment step for improving biodegradability and reducing acute ecotoxicity. The resulting byproducts can be removed through subsequent biological treatments. Total mineralization through oxidation processes is highly expensive, whereas a combination of oxidation processes and biological options could be a cheaper method for the degradation of total organics. Table 1 presents the findings from several studies of the ozonation of wastewater containing azo dye alone [10–16]. The present study was designed to demonstrate the decolorization and enhanced biodegradability of synthetic batik wastewater containing Reactive Black 5 (RB5) and Reactive Orange 16 (RO16) after semi-batch ozonation treatment. RB5 and RO16 were used, respectively, as model diazo and monoazo dyes.

**Table 1.** Studies of ozonation of RB5 and RO16.

| Dye | Ozonation Conditions | Initial pH and Temperature | Removal | Ref. |
|---|---|---|---|---|
| RB5 2 g/L, 1.2 L | 20.5 mg/L, 20 L/h, 6 h | pH 6.1 | COD 40%, TOC 25% | [10] |
| RB5, 1500 mg/L, 500 mL | Initial 55.5 mg/L and 1.5 L/min, 25 min | pH 10 | COD 50%, Dye 94% | [11] |
| RB5, 1500 mg/L, 1.157 L | 5 g/h, 10 min | pH 10.13 | Color 70%, COD 50% | [12] |
| RB5, 100 mg/L, 2 L | 40.88 mg/min | 20 °C | Color 96.9% (5 h), COD 77.5% (2 h) | [14] |
| RB5, 100 mg/L, 250 mL | 5 g/L, 40 min | pH 2–10, 25 °C | Dye 99.9%, COD 100% | [13] |
| RO16, 90 mg/L, 500 mL | 20–80 g/m$^3$, 400 mL/min, 5–17 min | pH 7 (2,7,11), 20 °C | Dye > 90% | [15] |
| RO16, 25–100 mg/L, 2 L | 51 mg/L, 5 min | pH 6.2–7.8 | Color 97%, TOC 48% | [16] |
| RO16, 100 mg/L, 250 mL | 5 g/L, 70 min | pH 2–10, 25 °C | Dye 99%, COD 100% | [13] |
| Mixture of RB5 and RO16, 48 mg/L each, 2 L | 7.3 g/m$^3$, 4 L/min, 15–30 min | pH 12, 25 °C | Color > 99% (10 min); RB5, RO16 > 99% (5 min); COD 27%; TOC 34% (15 min) | This study |
| Mixture of RB5 and RO16, 96 mg/L each, 2 L | | | Color > 98% (30 min), TOC 4.3% (30 min) | |

## 2. Materials and Methods

### 2.1. Synthetic Batik Wastewater

The synthetic batik wastewater contained 48 mg/L RB5, 48 mg/L RO16, and 3 g/L sodium silicate. The pH was 12 because of the alkaline characteristics of the silicate. Both RB5 ($C_{26}H_{21}N_5Na_4O_{19}S_6$) and RO16 ($C_{20}H_{17}N_3Na_2O_{11}S_3$) were purchased from

Sigma Aldrich (St. Louis, MO, USA). The high-strength synthetic batik wastewater for an additional experiment contained 96 mg/L RB5, 96 mg/L RO16, and 3 g/L sodium silicate.

### 2.2. Ozonation Tests

A schematic diagram of the ozonation process is presented in Figure 1. A 2 L glass reactor was filled with 2.0 L synthetic batik wastewater. Ozone gas was generated using a pressure swing adsorption oxygen generator (PSA ITO-04-01; IBS Inc., Suita, Japan) and an ozone generator (ED-OG-S3; EcoDesign, Inc., Hiki, Japan). Batik wastewater was treated with ozone gas (7.3 g/m$^3$, 4 L/min) in a semi-batch mode at 25 °C. As a control experiment, pure water was also treated with ozone gas. The ozone concentration in the exhaust gas was monitored using a UV ozone monitor (ED-OA-1; EcoDesign, Inc.). The residual ozone in the exhausted gas was treated using an ozone decomposer (ED-MD9-500S; EcoDesign, Inc., Hiki, Japan). Water samples were collected from the sampling port of the reactor.

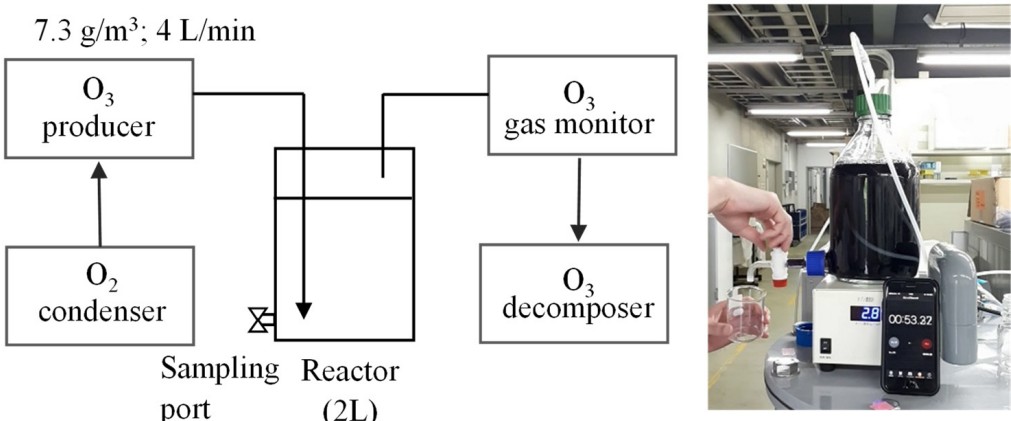

**Figure 1.** Schematic of the ozonation reactor used for synthetic batik wastewater.

The decay of the RB5, RO16, and color unit (CU) were fitted to the first-order kinetic model as

$$C = C_0 \exp(-kt) \tag{1}$$

where $C$ represents the concentration of RB5, RO16, or CU; $C_0$ is the initial concentration; $k$ is the decay rate constant (min$^{-1}$); and $t$ is the ozonation time (min).

### 2.3. Biodegradation Tests

Each 420 mL sample of untreated or ozonation-treated wastewater was processed by activated sludge in duplicate using a respirometer (OxiTop®; WTW GmbH, Weilheim, Germany) under dark conditions at 25 °C for 7 days. The pH value of the wastewater samples was adjusted to 7.6 by adding a HCl solution. To each test bottle, activated sludge (final concentration 30 mg/L) and nutrients were added according to the protocol provided in the Organization for Economic Co-operation and Development 301F (OECD 301F, Manometric Respirometry Test) [17]. The final nutrient concentrations were as follows: $K_2HPO_4$ 21.8 mg/L, $KH_2PO_4$ 8.5 mg/L, $Na_2HPO_4 \bullet 12H_2O$ 44.6 mg/L, $NH_4Cl$ 1.7 mg/L, $MgSO_4 \bullet 7H_2O$ 22.5 mg/L, $CaCl_2$ 27.5 mg/L, and $FeCl_3 \bullet 6H_2O$ 0.25 mg/L. A control experiment was also conducted using blank water containing activated-sludge microorganisms and nutrients, without synthetic dyes. The activated sludge was collected from a master culture reactor operated in fill-and-draw mode and fed with synthetic domestic wastewater [18]. The BOD value was calculated from the difference in dissolved oxygen (DO) consumption between wastewater and blank water.

### 2.4. Analytical Procedures

For this study, the CU was determined according to a combination of the platinum cobalt method and the measurement of the absorbance at 465 nm. The absorbance spectrum was measured using a spectrometer (DPM-MTSP; Kyoritsu Chemical-Check Lab. Corp., Yokohama, Japan). The wavelengths of maximum absorbance were 600 nm and 495 nm, respectively, for RB5 and RO16. The relation between the absorbance and dye concentration was found to be linear in a preliminary study.

Three-dimensional excitation emission matrix (EEM) fluorescence spectroscopy analysis was used to characterize five- or ten-fold diluted wastewater using a spectrophotometer (F-7000; Hitachi High-Tech Corp., Tokyo, Japan). The photomultiplier voltage was 700 V. The scanning range, scan width, and slit width were, respectively, 200–600 nm, 10 nm, and 5 nm for both excitation and fluorescence. The scanning speed was 30,000 nm/min.

The $COD_{Cr}$ value was measured using the reactor digestion method with a colorimeter (DR890; HACH Co., Loveland, CO, USA). The total organic carbon (TOC) value was measured using a TOC analyzer (TOC-V; Shimadzu Corp., Kyoto, Japan). Dissolved ozone was measured using a polarography sensor (OZ-221AA; DKK-TOA Corp., Tokyo, Japan).

## 3. Results

### 3.1. Decolorization of Synthetic Batik Wastewater by Ozonation

The synthetic batik wastewater containing 48 mg/L RB5 and 48 mg/L RO16 was dark purple, but the wastewater became light orange-yellow after 2–5 min ozonation, eventually becoming clear after 15 min ozonation, as shown in Figure 2 and Video S1. Ozonation reduced the CU from 4,240 to 70 in 10 min and further to below 50 in 15 min. Figure 3 shows the decreased absorbance of wastewater at 495 nm for RO16 and 600 nm for RB5 during ozonation. The first-order decay constant ($k$) for 5 min was found to be 1.11 min$^{-1}$ for RB5 ($r^2 = 0.99$), 0.82 min$^{-1}$ for RO16 ($r^2 = 0.99$), and 0.69 min$^{-1}$ for CU ($r^2 = 0.99$), suggesting the higher degradability of RB5.

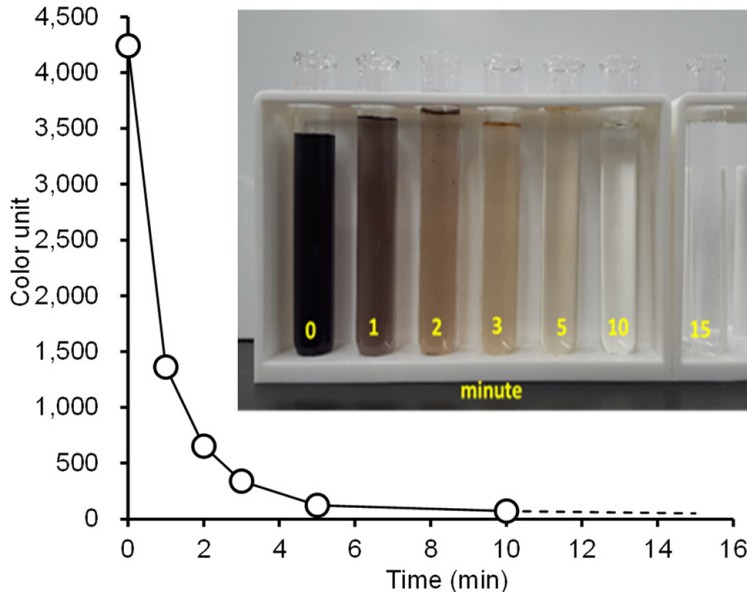

**Figure 2.** Decrease in the color unit (CU) of synthetic batik wastewater containing 48 mg/L RB5 and 48 mg/L RO16 after ozonation at 25 °C and pH 12. CU at 15 min was below the detection limit of 50.

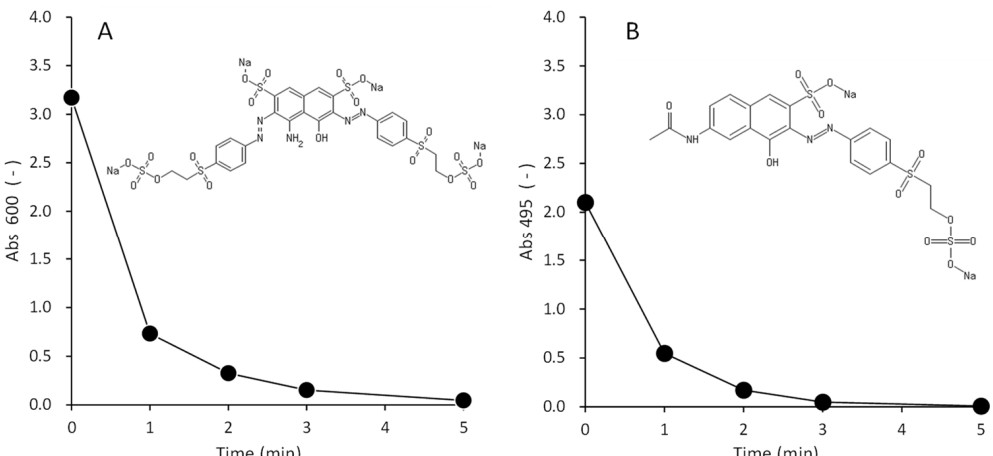

**Figure 3.** Absorbance at 495 nm and 600 nm of synthetic batik wastewater containing 48 mg/L RB5 and 48 mg/L RO16 during ozonation at 25 °C and pH 12. A600 for RB5 (**A**) and A495 for RO16 (**B**). The absorbance of wastewater treated by ozonation for 10–15 min was less than 0.01.

Figure 4 shows the ozone concentration in the gas exhausted from the reactor. During the 15 min experimental period, the amount of ozone that inflowed into the 2.0 L reactor was calculated to be 438 mg $O_3$ according to the gas concentration (7.3 g $O_3/m^3$) and the flow rate (4 L/min), resulting in an ozone injection ratio of 219 mg $O_3/L$. In a control experiment treating pure water, the ozone concentration reached about 7.3 g/$m^3$ in the exhausted gas within a few minutes. In the wastewater treatment experiment, the ozone concentration increased gradually to about 6.8 g $O_3/m^3$ in the exhausted gas. The dissolved ozone concentration in the water phase was almost zero. The difference in the amount of exhaust ozone gas between the control and the treatment experiments suggested that the ozone consumed by oxidizing the dyes was 113 mg $O_3$, corresponding to 25.8% of the inflow amount.

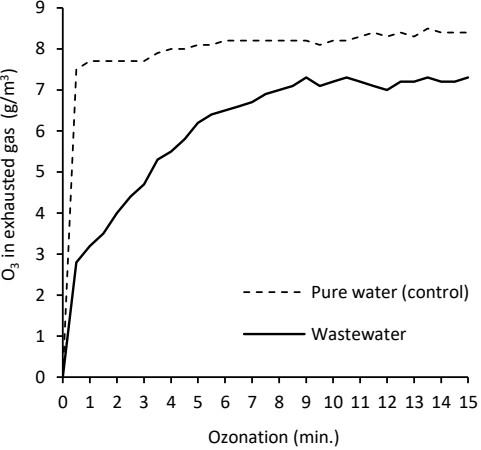

**Figure 4.** Ozone concentration in the gas exhausted from the reactor for synthetic batik wastewater containing 48 mg/L RB5 and 48 mg/L RO16 at 25 °C.

### 3.2. Biodegradation of Synthetic Batik Wastewater

Figure 5 shows the DO consumption by activated-sludge microorganisms in synthetic batik wastewater. Microorganisms consumed less oxygen in wastewater before ozonation than in blank water, indicating the inhibition of microbial respiration by the toxicity of the azo dyes. Therefore, the BOD of synthetic batik wastewater could not be estimated. Nevertheless, the microorganisms in the wastewater after 15 min ozonation consumed more DO than in the blank wastewater. The results indicated that ozonation decreased the toxicity and increased the biodegradability of organic matter in the synthetic batik wastewater.

The BOD$_5$ and BOD$_{14}$ for the wastewater after 15 min ozonation were determined to be, respectively, 5.2 mg/L and 9.9 mg/L.

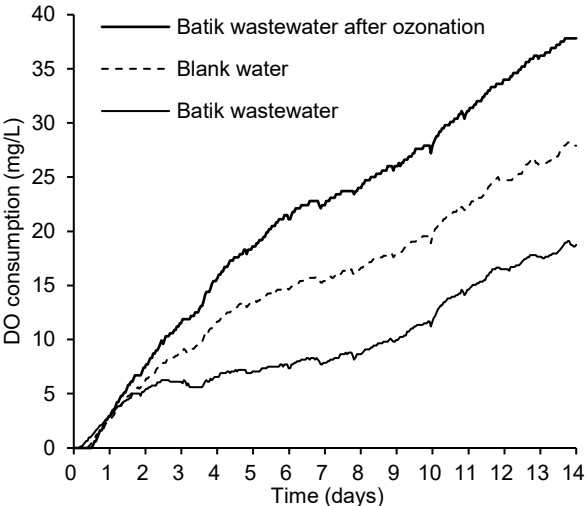

**Figure 5.** DO consumption by activated-sludge microorganisms at 25 °C in synthetic batik wastewater containing 48 mg/L RB5 and 48 mg/L RO16 before and after 15 min ozonation.

### 3.3. Evaluation of Total Ozonation and Biodegradation

The changes in the TOC and COD concentrations in the synthetic batik wastewater are shown in Figure 6. The TOC concentration was reduced from 19.9 mg/L to 19.4 mg/L by biodegradation alone, 13.1 mg/L by ozonation alone, and 10.2 mg/L by ozonation followed by biodegradation. The COD concentration was reduced from 86 mg/L to 73 mg/L by biodegradation alone, 63 mg/L by ozonation alone, and 54 mg/L by ozonation followed by biodegradation. The BOD$_5$/COD and BOD$_5$/TOC ratios of synthetic batik wastewater after 15 min ozonation were, respectively, 0.083 and 0.39. These are indicators of the proportion of biodegradable organic matter to total organic matter. The results confirmed that biological treatment alone was ineffective for reactive dye degradation. Ozonation was effective for decolorization, but byproducts remained in the wastewater.

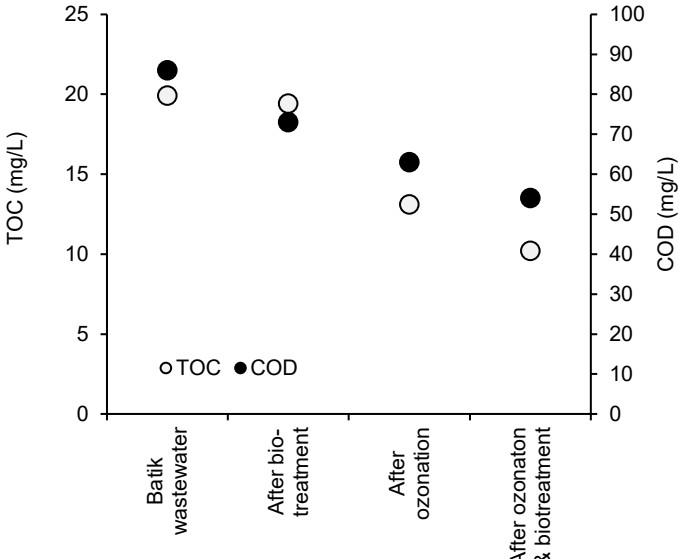

**Figure 6.** Decrease in TOC and COD concentrations in synthetic batik wastewater containing 48 mg/L RB5 and 48 mg/L RO16 after 14 days of biodegradation; 15 min ozonation; and 15 min ozonation followed by 14 days of biodegradation.

Figure 7 shows the 3D EEM spectra of the synthetic batik wastewater. The peak of RO16 at Ex/Em = 240/425–450 was predominant over the peak of RB5 at Ex/Em = 285/415 nm in batik wastewater (Figure 7A). These peaks remained in wastewater after biodegradation alone (Figure 7F). Although the peaks of the batik wastewater decreased after 15 min ozonation (Figure 7D), small peaks appeared temporarily at Ex/Em = 260/360 and 320/420 after 1 min (Figure 7B) and 3 min ozonation (Figure 7C), respectively, suggesting the accumulation of ozonolysis products. After 15 min ozonation followed by 14 days of biodegradation, these peaks disappeared from wastewater (Figure 7E).

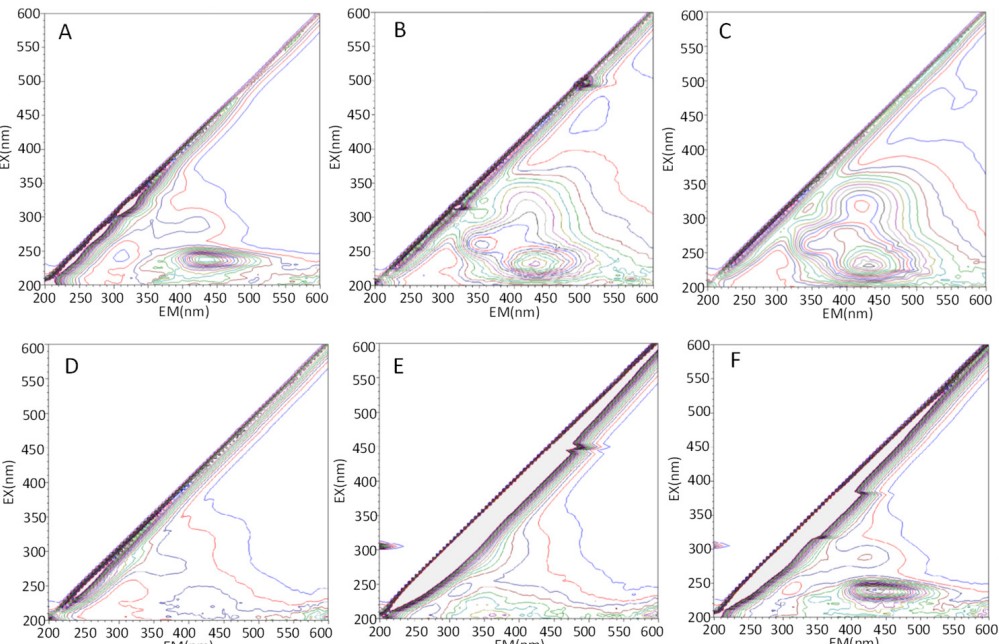

**Figure 7.** 3D EEM spectra for synthetic batik wastewater containing 48 mg/L RB5 and 48 mg/L RO16 treated by ozonation for 0 min (**A**), 1 min (**B**), 3 min (**C**), and 15 min (**D**); ozonation for 15 min followed by biodegradation for 14 days (**E**); and biodegradation for 14 days (**F**). All samples were 5-fold diluted for analysis.

### 3.4. Ozonation and Biodegradation Tests for High-Strength Synthetic Batik Wastewater

Additional experiments were conducted to investigate the ozonation and biodegradation of high-strength synthetic batik wastewater containing 96 mg RB5 and 96 mg/L RO16. As shown in Figure 8, the dark purple wastewater became light orange-yellow after 5 min ozonation, eventually becoming clear after 20–30-min ozonation. Ozonation reduced the CU from 9500 to 1700 in 5 min and further to 143 in 30 min. The first-order decay constant ($k$) for 15 min was determined to be 0.23 min$^{-1}$ for CU ($r^2$ = 0.98).

As shown in Figure 9, the BOD of the high-strength synthetic batik wastewater could not be estimated because less oxygen was consumed than in blank water. The microorganisms in the wastewater after 30 min ozonation consumed more DO than in blank wastewater. The BOD$_5$ for the high-strength wastewater after 30 min ozonation was found to be 15.4 mg/L.

As shown in Figure 10, the TOC concentration in the high-strength wastewater was reduced from 52.6 mg/L to 39.6 mg/L after 14 days of biodegradation alone, 50.3 mg/L after 30 min ozonation alone, and 17.6 mg/L after 30 min ozonation followed by 14 days of biodegradation. The BOD$_5$/TOC ratio of the high-strength synthetic batik wastewater after 30 min ozonation was 0.31.

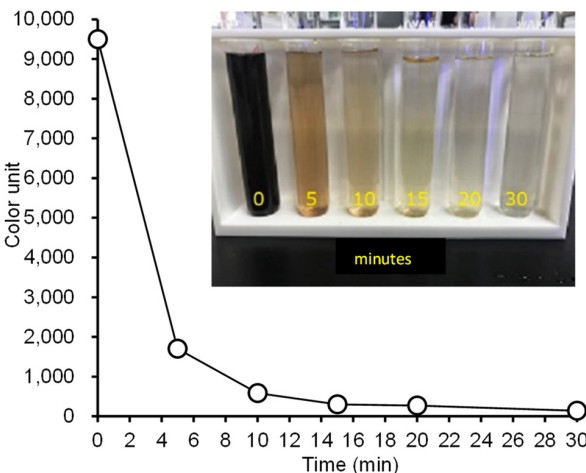

**Figure 8.** Decrease in the color unit (CU) of synthetic batik wastewater containing 96 mg/L RB5 and 96 mg/L RO16 during ozonation at 25 °C and pH 12.

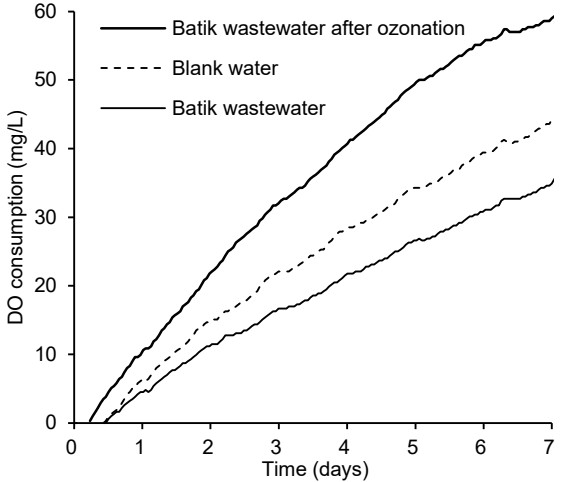

**Figure 9.** DO consumption by activated-sludge microorganisms at 25 °C in high-strength synthetic batik wastewater containing 96 mg/L RB5 and 96 mg/L RO16 before and after 30 min ozonation.

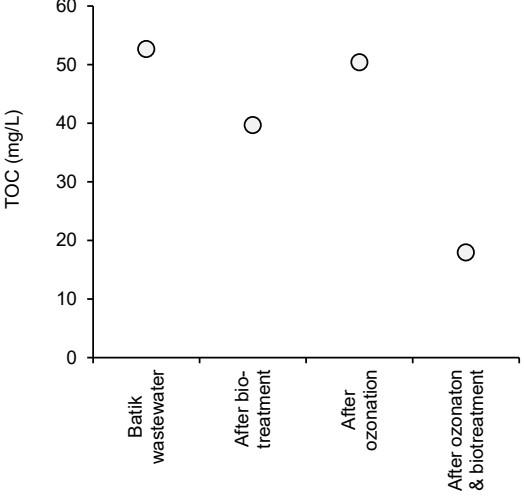

**Figure 10.** Decrease in TOC concentration in synthetic batik wastewater containing 96 mg/L RB5 and 96 mg/L RO16 after 14 days of biodegradation, 30 min ozonation, and 30 min ozonation followed by 14 days of biodegradation.

Figure 11 shows the 3D EEM spectra of the high-strength synthetic batik wastewater. The peaks of RO16 at Ex/Em = 240/425–450 and Ex/Em = 230/350 were predominant (Figure 11A). These peaks decreased after 30 min ozonation (Figure 11B). After 30 min ozonation followed by 14 days of biodegradation, a small peak at Ex/Em = 280/370 remained in the treated wastewater (Figure 11C).

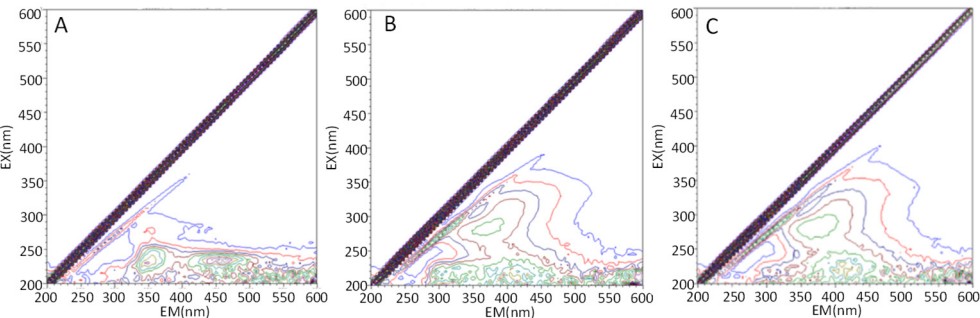

**Figure 11.** 3D EEM spectra for high-strength synthetic batik wastewater containing 96 mg/L RB5 and 96 mg/L RO16 treated by ozonation for 0 min (**A**) and 30 min (**B**); and ozonation for 30 min followed by biodegradation for 14 days (**C**). All samples were 10-fold diluted for analysis.

## 4. Discussion

Synthetic batik wastewater typically containing 16–2000 mg/L of dyes (Table 1), silica, wax, and resin has been used to study alternative treatment systems [3–5]. Rashidi et al. [3] studied the use of a baffle separation tank for removing wax from synthetic wastewater (reactive dye 16 mg/L, sodium silicate 1 g/L, and wax 7.7 g/L). Although 92–95% of the wax and 32–42% of the silicate were removed, only 2–5% of the reactive dyes were removed by baffle separation [3]. Therefore, this study specifically examined the removal of azo dyes (48–96 mg/L) from batik wastewater.

Ozone and hydroxyl radicals under alkaline conditions are highly reactive with the azo double bond (-N=N-). The ozonation process for RB5 and RO16 in individual solutions has previously been studied, as shown in Table 1. This study demonstrated the decolorization of RB5 and RO16 in a mixture at pH 12 (Figures 2 and 8), suggesting the higher degradability of RB5 according to the first-order decay constant (1.11 min$^{-1}$ for RB5, 0.82 min$^{-1}$ for RO16) (Figure 3). The first-order decay constant for the CU of high-strength wastewater (0.23 min$^{-1}$, 96 mg/L RB5 and RO16) was lower than that of standard wastewater (0.69 min$^{-1}$, 48 mg/L RB5 and RO16). Onder et al. [13] reported first-order decay constants of 0.1918 min$^{-1}$ for RB5 (100 mg/L, pH 2) and 0.0794 min$^{-1}$ for RO16 (100 mg/L, pH 6). Under acidic conditions, the oxidation process of azo dye is largely attributed to the molecular ozone contribution, but under basic conditions, contributions from both free radicals and molecular ozone are observed, thereby yielding an overall increase in the dye decay rate [19]. Only a small portion of the injected ozone was consumed for decolorization (25.8%, Figure 4), because millimeter-order ozone bubbles were used in this study. Our previous study demonstrated the efficient consumption of the injected ozone (82%) for humic acid decolorization using ultrafine ozone bubbles (0.05–20 micrometers) [20]. Further studies of the optimization of the ozonation process (according to the bubble size, concentration, flow rate, and reaction time) must be conducted to achieve efficient batik wastewater treatment.

Although the degradation processes of RB5 and RO16 were observed using 3D EEM (Figures 7 and 11), the ozonolysis products were not analyzed directly in this study. A few reports have described the degradation pathways of these dyes [11,15]. Tizaoui and Grima [15] proposed degradation pathways for RO16 via ozonation. The azo bond breakage of RO16 possibly engenders the formation of (2-(4-nitrosophenyl)sulfonylethyl hydrogen sulfate, 6-acetamido-4hydroxy-3-nitroso-naphtalene-2-sulfonic acid, and 6-acetamido-4-hydroxy-3-aminonaphthalene-2-sulfonic acid. In fact, the degradation products generated by the ozonation of RO16 were nitrosobenzene; nitrobenzene; benzene-1,4-diol;

1,4-benzoquinone; acetamide; and phthalic, maleic, oxalic, acetic, and formic acid [15]. Using ion chromatography and UV–vis spectroscopy, Venkatesh et al. [11] detected oxalate, sulfate, nitrate, and chloride during the ozonation of RB5.

This study assessed the toxicity of the mixture of RB5 and RO16 for microbial respiration and the detoxification of the dyes by ozonation (Figures 5 and 9). The BOD of the synthetic batik wastewater could not be estimated because of its toxicity. The $BOD_5/COD$ and $BOD_5/TOC$ ratios of the synthetic batik wastewater after 15–30 min ozonation were 0.085 and 0.31–0.39, respectively. Suryawan et al. [14] also reported that ozonation drastically increased the $BOD_5/COD$ ratio of RB5 solution from 0.27–0.38 to 0.40–0.65. It is generally suitable to use a biological treatment process when the $BOD_5/COD$ ratio of wastewater is higher than 0.5 [21]. Activated-sludge microorganisms removed a portion of these byproducts of ozonation, although some metabolites might have remained in the treated water. Wang et al. [10] reported that the first byproducts generated by the partial ozonation of RB5 were more biodegradable than the parent chemical, but that subsequent oxidation produced toxic intermediates. Venkatesh et al. [11] investigated the ozonation of RB5 followed by processing in an anaerobic bioreactor. They achieved around 70% decolorization and 50% COD removal after 10 min ozonation. The system of ozonation and anaerobic treatment showed a COD removal of 90% and a dye removal of 94%. Castro et al. [16] investigated the ozonation of 25–100 mg/L RO16 followed by processing in an aerobic bioreactor. Although the maximum TOC removal from the RO16 solution by ozonation was only 48%, the ozonolysis products did not affect the subsequent bioreactor performance, which achieved an average COD removal of 93%. The activated-sludge microorganisms used for this study had been cultivated with synthetic domestic wastewater. Microorganisms acclimated to batik wastewater after ozonation are necessary for further studies of the complete removal of byproducts. The existing WWTPs in Pekalongan city use conventional anaerobic and aerobic bioprocesses [6]. These WWTPs can be upgraded to achieve robust treatment by installing the ozonation process as a pretreatment.

## 5. Conclusions

Ozonation efficiently decolorized and enhanced the biodegradability of synthetic batik wastewater containing RB5 and RO16. Although it requires large amounts of electricity, ozonation could be an efficient process for improving the biodegradability of azo dyes, as well as reducing their acute ecotoxicity. However, some chemicals might remain in water treated by ozonation and subsequent biodegradation. Further studies on the optimization of the ozonation process (in terms of the bubble size, gas concentration, flow rate, and reaction time) and the acclimation of microorganisms to ozonized wastewater must be conducted to achieve efficient wastewater treatment for a sustainable batik industry.

**Supplementary Materials:** The following supporting information can be downloaded at: https://www.mdpi.com/article/10.3390/w14203330/s1, Video S1: Decolorization of synthetic batik wastewater by ozonation.

**Author Contributions:** Conceptualization, A.P. and S.S.; methodology, T.S. and S.S.; software, A.P., T.S., S.G. and S.S.; validation, A.P., T.S. and S.S.; formal analysis, A.P. and T.S.; investigation, A.P., T.S. and S.G.; resources, T.S. and S.S.; data curation, A.P. and T.S.; writing—original draft preparation, A.P. and S.G.; writing—review and editing, S.S.; visualization, A.P. and S.S.; supervision, T.A.A. and S.S.; project administration, T.A.A. and S.S.; funding acquisition, A.P. and S.S. All authors have read and agreed to the published version of the manuscript.

**Funding:** This study was partly supported by the Indonesian Linkage Master Program under the fourth Professional Human Resource Development Project (PHRDP-IV) funded by the Japan International Agency (JICA), Ritsumeikan Advanced Research Academy (RARA), and the Obayashi Foundation.

**Institutional Review Board Statement:** Not applicable.

**Informed Consent Statement:** Not applicable.

**Data Availability Statement:** Not applicable.

**Conflicts of Interest:** The authors declare no conflict of interest.

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
