# Peer review of "Decolorization and Biodegradability Enhancement of Synthetic Batik Wastewater Containing Reactive Black 5 and Reactive Orange 16 by Ozonation"

_water, doi:10.3390/w14203330_

Round 1
Reviewer 1 Report (Previous Reviewer 2)
No comments.
Author Response
Thank you for reviewing our manuscript.

Reviewer 2 Report (New Reviewer)
The presented manuscript includes the study of decolorization and biodegradability enhancement of synthetic batik wastewater containing Reactive Black 5 and Reactive Orange 16 by ozonation.
The Paper is of interest. Of course, the paper would be looking better with such variables as ozone concentration in a gas mixture, volume flow of ozone-gas mixture, and testing at the end on real wastewater. Without such experiments, it looks quite simple.
1. Fig. 7 and 11. Please, make the bigger(readable) font on all X and Y- axis
Author Response
We appreciate you for reviewing our paper.
- 7 and 11. Please, make the bigger(readable) font on all X and Y- axis.
For Figs. 7 and 11, font on all X and Y- axis were made bigger.

This manuscript is a resubmission of an earlier submission. The following is a list of the peer review reports and author responses from that submission.
Round 1
Reviewer 1 Report
The manuscript is well written and organized, but without suficcient experimental content. Besides performing complementary experiments before resubmiting the manuscript, I would like to make the following suggestions:
Page 2, line 57: “Ozonation is an advanced oxidation process (AOP) used for treating recal- 56 citrant organics by the generation of hydroxyl radicals (•OH)”. This is not totally correct. Ozone is only transformed in hydroxyl radicals under certain conditions. Ozone can react either by direct oxidation of organic pollutants (mostly at acidic conditions), or via hydroxyl radical formation (mainly produced under alkaline conditions)
Materials and methods.
Page 2, lines 73-74: “The synthetic batik wastewater contained 48 mg/L RB5, 48 mg/L RO16, and 3 g/L sodium silicate.” Please explain why these concentrations of the model azo dyes were chosen. Are they representative of real concentration in real wastewaters? This justification must be included in the manuscript.
Page 2, lines 82-83: “Batik wastewater was treated with ozone gas (7.3 g/m3 , 4 L/min) in a semi-batch mode at 25°C.” Please explain why these ozonation conditions were chosen. Taking into account other investigations dealing with ozonation, the gas flow rate seems too high. Did the authors try other ozone concentrations and gas flows? This would be an interesting contribute to your investigation.
Results:
Page 4, lines 140-141: “The difference of the exhaust ozone gas amount between the control and treatment experiments suggests ozone consumption for oxidizing the dyes: 113 mg-O3, corresponding to 25.8% of the inflow amount.” The authors here have the evidence that only a small portion of the ozone in the gas phase is consumed, so it would be interesting to make more experiments with lower ozone concentrations (for costs reductions) and verify if they result in the same efficiency.
Page 5, line 160: The BOD to COD and TOC rations should be better explained. The values obtained by the authors have to be compared to some standards. For example, if the BOD to COD ration is below x, it means that the treated water presents low biodegradability. This kind of explanations are missing for both ratios.
Conclusions: This study was conducted with very few experiments, exploring only one experimental condition. More experiments exploring other ozonation conditions should be performed for to support this investigation. Other complementary studies without experimental content could also be performed, such as economic analysis to confirm that coupling ozone after biological treatment allows reducing operational costs.
Reviewer 2 Report
93: Please complete the data on the methodology of the biodegradation test in the text place where reference is made to the OECD protocol (98-99): amounts, types of substances added to Oxitop bottles. The specific number and name of the OECD test used should also be given.
Under Figures 5 and 6 complete the signature with the data on the ozonation time.